# Exploring the Muscle-to-Fat Ratio of Pediatric Patients with Thyroid Disorders and Its Interaction with Thyroid Function and Metabolic Syndrome Components

**DOI:** 10.3390/jcm14041255

**Published:** 2025-02-14

**Authors:** Avivit Brener, Yuval Stark, Gal Friedman Miron, Shay Averbuch, Erella Elkon-Tamir, Ophir Borger, Yael Lebenthal

**Affiliations:** 1The Institute of Pediatric Endocrinology, Diabetes and Metabolism, Dana-Dwek Children’s Hospital, Tel Aviv Sourasky Medical Center, Tel Aviv 6423906, Israel; avivitb@tlvmc.gov.il (A.B.); galf26@gmail.com (G.F.M.); elladavy@yahoo.com (E.E.-T.); ophir.bor@gmail.com (O.B.); 2Faculty of Medicine, Tel Aviv University, Tel Aviv 6997801, Israel; yuvalstark@gmail.com (Y.S.); shayaverbuch97@gmail.com (S.A.); 3The Nutrition and Dietetics Unit, Tel Aviv Sourasky Medical Center, Tel Aviv 6423906, Israel

**Keywords:** body composition, children and adolescents, lipid profile, muscle-to-fat ratio (MFR), overweight/obese, thyroid function

## Abstract

**Background/Objectives:** The standard evaluation of children and adolescents suspected of having thyroid disorders consists of anthropometric measurements. Body composition features provide additional information for enhanced therapeutic management. We explored the muscle-to-fat ratio of pediatric patients referred for thyroid disorders and its interaction with thyroid function and metabolic syndrome components. **Methods**: This retrospective cross-sectional study consisted of 147 pediatric subjects (ages 5–19 years) diagnosed with childhood-onset thyroid disorders treated at a tertiary medical center. Sociodemographic, clinical and laboratory data [thyroid-stimulating hormone (TSH), free T4 (FT4), and lipid profile] were extracted from the electronic medical records. Body composition was measured using bioimpedance analysis (Tanita MC-780 MA and GMON Professional Software). Body mass index (BMI), appendicular muscle mass (ASMM), and muscle-to-fat ratio (MFR) were converted to z-scores. **Results**: The diagnoses included Hashimoto thyroiditis (30.6%), subclinical hypothyroidism (26.5%), congenital hypothyroidism (21.7%), and Graves’ disease (21%). Based on BMI z-scores, 31.3% of the cohort was overweight or obese. The TSH levels were positively correlated with the BMI z-scores (r = 0.238, *p* = 0.005) and negatively with the MFR z-scores (r = 0.215, *p* = 0.012). The ASMM z-scores were negatively associated with the FT4 levels (r = −0.255, *p* = 0.003). Dyslipidemia was prevalent. TSH was correlated with LDL cholesterol (r = 0.472, *p* < 0.001) and triglycerides (r = 0.232, *p* = 0.05). **Conclusions**: Elevated thyroid-stimulating levels were linked to higher BMI and lower MFR levels. Our findings on the relationship between thyroid function and lipid profile underscore the necessity of optimizing thyroid balance and implementing targeted lifestyle interventions to improve body composition in young patients with thyroid disorders.

## 1. Introduction

Normal thyroid function is essential for neurocognitive function, growth, and metabolic regulation during childhood and adolescence [1]. Thyroid hormones, primarily thyroxine (T4) and triiodothyronine (T3), produced by the thyroid gland’s follicular cells, influence nearly every cell in the body via intranuclear receptors. Thyroid disorders, including congenital hypothyroidism and autoimmune thyroid diseases (Hashimoto thyroiditis and Graves’ disease), when not appropriately treated, can disrupt growth and metabolic processes [1]. Thyroid hormones influence weight status, body composition, and lipid metabolism [2]. A negative association between free T3 (FT3) and body weight supports the role of T3 as a regulator of the resting metabolic rate and energy expenditure [3]. While BMI is a common anthropometric measure, it does not accurately reflect muscle and fat distribution. There are several reports that suggest correlations between thyroid hormone levels (T3, T4, and TSH) and body composition, with potential links to mitochondrial dysfunction [3]. Body weight and fat mass were associated with thyroid hormone concentrations in patients with both hypothyroidism and hyperthyroidism [4]. In men without thyroid disorders, thyroid hormone levels were associated with adiposity indices but not with TSH [5].

Thyroid hormones determine cardiovascular risk through their effect on blood pressure and lipid metabolism. By acting on myocardial and vascular endothelial tissues, they induce vasodilation, reduce systemic vascular resistance, and activate the renin–angiotensin–aldosterone system, thereby increasing the plasma volume and cardiac output [6]. Hypothyroidism typically presents with hypercholesterolemia and elevated low-density lipoprotein (LDL) levels, while hyperthyroidism is associated with lower serum total cholesterol, LDL, and apolipoprotein B levels [7,8]. These metabolic consequences underlie the importance of thyroid hormones in overall cardiovascular health and metabolic regulation.

In January 2018, our Institute of Pediatric Endocrinology implemented the analysis of body composition by means of bioimpedance analysis (BIA) as an addition to the standard intake assessment of subjects aged 5 years and older referred for endocrine consultation [9]. The BIA tool for assessing body composition provides a more nuanced determination of the subject’s weight status and overall health. In this study, we explored the body composition components and examined their interaction with thyroid function and metabolic outcomes in children and adolescents treated and followed for various thyroid disorders.

## 2. Materials and Methods

### 2.1. Study Design

This single-center, cross-sectional retrospective study included children and adolescents (5–19 years) with the diagnosis of thyroid disorders who had been treated at a tertiary medical center between January 2018 and June 2022. The study participants were identified through a search of the institutional bioelectrical impedance analysis (BIA) database for the following diagnostic terms: ’congenital hypothyroidism’, ’Hashimoto thyroiditis’, ’Graves’ disease’, and ’subclinical hypothyroidism’.

Patients with congenital hypothyroidism were diagnosed by confirmatory testing following detection by the National Newborn Thyroid Screen universally obtained at 36–48 h of age [10]. Screening preterm infants for congenital hypothyroidism typically involved evaluating their thyroid function at 2 weeks of age, comparing the results to established reference ranges, and conducting further testing if necessary to confirm the diagnosis [10]. Hashimoto thyroiditis was diagnosed in cases with elevated TSH levels and low levels of FT4, coupled with increased antithyroid peroxidase (TPO) and/or anti-thyroglobulin auto-antibodies [11]. Subclinical hypothyroidism was defined as the sole elevation of TSH levels above the normal range for the age [12,13] in the presence of normal FT4 levels and absence of clinical manifestations. Graves’ disease was diagnosed as depressed TSH levels coupled with elevated FT4 levels and evidence of thyroid-stimulating immunoglobulins [14]. All patients with hypo- or hyperthyroidism were treated to achieve clinical and laboratory euthyroid condition. The laboratory results were interpreted according to age-appropriate reference ranges (provided by the manufacturer) as follows: TSH (mIU/mL): from birth to 1 month = 0.64–10; from 1 to 6 months = 0.64–7; from 6 months to 1 year = 0.64–6.27; from 1 to 18 years = 0.51–4.94; from 18 years onwards = 0.40–4.70. FT4 (ng/dL): from birth to 1 month = 0.66–2.37; 1 month to 1 year = 0.71–1.97; from 1 year onwards: 0.70–1.80.

The search yielded 489 reports pertaining to 221 patients with thyroid disorders, and their BIA data were linked to their electronic medical records. Excluded were 74 patients with medical conditions that may affect body composition and/or cardiometabolic health: type 1 diabetes, celiac disease, lipodystrophy, genetic syndromes, malignancies, metabolic bone disease, steroid therapy, medical conditions that may lead to fluid retention, and gender-affirming hormonal treatment due to gender diversity. Included in the study were 147 patients aged 5 to 19 years diagnosed with the aforementioned thyroid disorders. They all had blood pressure measurements and laboratory evaluations of thyroid function conducted in close proximity to their BIA assessment. The study population flowchart is presented in Figure 1.

### 2.2. Ethics

The study protocol was approved by the ethics committee of the Tel Aviv Sourasky Medical Center, which waived the need for informed parental consent (no. 0282-22-TLV). The data were handled in accordance with the principles of good clinical practice.

### 2.3. Data Collection

The routine clinical management of patients with thyroid disorders at our center involves biannual or triannual clinic visits, during which evaluations include a medical interview (anamnesis) of patients and their parents, anthropometric assessment of height and weight, vital signs, physical examination, and review of the laboratory findings. In patients requiring medical intervention [e.g., thyroid hormone replacement for hypothyroidism and antithyroid medications (methimazole) for Graves’ disease], proper dosing relied on regular monitoring of growth and assessment of thyroid function. The clinical, anthropometric, and laboratory data that were collected at the time of the BIA assessment included sex; age; sociodemographic characteristics; medical history (including perinatal characteristics, medications, and medical conditions); blood pressure; anthropometric measurements; pubertal stage according to Marshall and Tanner [15,16]; thyroid function; and lipoprotein profile.

### 2.4. Body Composition Assessment

BIA was used to assess body composition in patients aged 5 years and older, utilizing the Tanita MC-780MA (Tanita, Tokyo, Japan) and GMON Professional Software (GMON Pro, Medizin and Service GmbH, Chemnitz, Germany) [17]. The BIA measures whole-body and segmental (trunk, upper, and lower limbs) fat and muscle. The calculated variables include ASMM (the sum of muscle mass of four limbs) and the MFR (appendicular skeletal muscle mass [kg]/fat mass [kg]). The z-scores for ASMM and MFR were calculated according to BIA pediatric reference curves [18].

### 2.5. Definition of Study Variables

The socioeconomic position (SEP) cluster and index were calculated by home address based upon the Israel Central Bureau of Statistics’ Classification. The residential SEP cluster was determined by the locality of residence, and it was scored on a scale of 1 to 10 and categorized into low (1–4), medium (5–7), and high (8–10) groups [19]. Height, weight, and BMI values were converted to z-scores according to the CDC 2000 growth charts [20]. The mid-parental height was calculated as follows: (paternal height [cm] + maternal height [cm] ± 13 cm)/2, and the mid-parental height z-scores were calculated [21]. The delta height z-score represented the patient’s linear growth compared to their potential genetic height, and it was calculated as the difference between the patient’s height z-score and the mid-parental height z-score. Pubertal stages were assessed using Tanner scores: genital stage 2 with testicular volume ≥ 4 mL in boys and breast bud appearance in girls marked onset of puberty, and Tanner stage 5 indicated full puberty [15,16].

Blood pressure (BP) measurements were conducted with an automated device (Welch Allyn—52000, Tycos Instruments, Inc., Arden, NC, USA) using the appropriate cuff for age according to American Heart Association guidelines [22]. Systolic BP and diastolic BP percentiles were calculated by means of an online age-based pediatric BP calculator [23]. Elevated BP was defined as the systolic and/or diastolic BP ≥90th percentile for sex, age, and height in patients <13 years [22]. Systolic and diastolic BP measurements in patients ≥13 years were presented as absolute values and also categorized according to the American College of Cardiology and American Heart Association [24] into normal (<120/80 mm Hg) or elevated (120–129/>80 mm Hg).

Dyslipidemia was defined by triglyceride levels (TG) ≥150 mg/dL, low high-density lipoprotein cholesterol (HDL-c) ≤40 mg/dL in boys and HDL-c ≤50 mg/dL in girls, and/or a dyslipidemia atherogenic index of TG-to-HDL-c ratio >2.

### 2.6. Statistical Analyses

All analyses were performed as two-sided using Statistical Package for the Social Sciences software version 28 (SPSS Inc., Chicago, IL, USA). The Kolmogorov–Smirnov test or the Shapiro–Wilk test was performed to test the normality of continuous data. Data were expressed as means ± standard deviations for normally distributed variables and median and interquartile range for skewed distribution. Body composition parameters (fat percentage, truncal fat percentage, ASMM, and MFR); BP (systolic and diastolic); thyroid function levels (TSH and FT4); and lipoprotein levels (TG and HDL-c) were analyzed as continuous variables. For quantitative data with normal distribution, the one-way ANOVA test was performed with the post hoc Mann–Whitney two-key test to compare between groups. For quantitative data with skewed distribution, an independent sample Kruskal–Wallis test was performed to compare between groups with adjustment by Bonferroni correction for multiple post hoc tests. For categorical data, the Chi-square test or Fischer exact test was performed for comparison. Sex, modes of conception and delivery, birthweight categories, and pubertal stage were presented and analyzed as categorical variables. The Chi-square test exact test was performed for categorical variables as appropriate. Spearman’s correlation coefficient statistical tests were applied to examine the correlation between two continuous data groups.

## 3. Results

The cohort comprised 147 children and adolescents with thyroid disorders: 45 (30.6%) were diagnosed with Hashimoto thyroiditis, 39 (26.5%) with subclinical hypothyroidism, 32 (21.7%) with congenital hypothyroidism, and 31 (21%) with Graves’ disease. A significant female predominance was shown in autoimmune thyroid disorders (60 girls out of the 76 patients with autoimmune thyroid disorders, *p* = 0.005). The socioeconomic position of the cohort was above average, with a median SEP cluster of 8 (range 1 to 10) and a median SEP index of 1.259 (range −1.806 to 2.802). Most of the pregnancies (92%) were conceived spontaneously, while assisted conceptions (by in vitro fertilization) were reported in 7.3% of cases, and one patient was adopted. Gestational diabetes was reported in 5.8% of the pregnancies. The majority of deliveries were spontaneous vaginal births (80.4%), while elective and urgent cesarean sections accounted for smaller percentages (15.7% and 3.9%, respectively). Most births occurred at term (80.5%). The median birth weight z-score for the entire cohort was −0.11 (range −2.81 to 2.14), with 6.1% born large for gestational age (LGA) and 6.1% born small for gestational age (SGA). Patients with congenital hypothyroidism tended to have lower birth weight z-scores (*p* = 0.061). The sociodemographic and perinatal characteristics of the cohort stratified by type of thyroid disorder are presented in Table 1.

The median age at BIA assessment was 13.3 years (range 5 to 19 years), with patients who had autoimmune thyroid disorders being significantly older (*p* < 0.001). The mean height z-score for the entire cohort was −0.01 ± 1.00 (range from −2.52 to 2.71), with 3.4% classified as having short stature (at or below the 3rd percentile for height). The genetic height potential for the entire cohort revealed a median delta height z-score of 0.07 (range −2.50 to 2.47), with no significant differences between the thyroid disorder groups. The median BMI z-score was 0.48 (range −3.27 to 2.32), with 46 patients (31.3%) classified as overweight/obese and 15 (10.2%) as underweight.

Body composition evaluation revealed a mean ASMM z-score of −0.36 ± 1.01 and a mean MFR z-score of −0.57 ± 1.02. The MFR z-scores were significantly higher in patients with Graves’ disease (*p* = 0.003). Thyroid function tests at BIA for the entire cohort revealed a median TSH of 4.11 mIU/L [IQR 1.61, 8.21] and median FT4 of 1.16 ng/dL [IQR 1.03, 1.36]. The TSH and FT4 levels differed significantly between groups at BIA, with patients with congenital hypothyroidism showing the highest TSH levels and those with subclinical hypothyroidism exhibiting the lowest FT4 levels (*p* < 0.001). At the BIA assessment, the lipid profile differed significantly across groups. Patients with Graves’ disease exhibited lower levels of total cholesterol, LDL-c, and non-HDL-c (*p* < 0.001), indicating a comparatively healthier lipid profile. Triglycerides and TG:HDL-c ratios were notably lower in patients with congenital hypothyroidism (*p* < 0.001), a finding likely influenced by the younger age of this group, where triglyceride levels are typically lower. Clinical and laboratory characteristics of the children and adolescents with thyroid disorders at the time of BIA evaluation are presented in Table 2.

The BMI z-scores were positively correlated with the TSH levels (r = 0.238, *p* = 0.005) but not with the FT4 levels (r = −0.165, *p* = 0.057). The ASMM z-scores were negatively correlated with the FT4 levels (r = −0.255, *p* = 0.003) but not with the TSH levels (r = 0.091, *p* = 0.291). The MFR z-scores were negatively correlated with the TSH levels (r = −0.215, *p* = 0.012) but not with the FT4 levels (r = 0.042, *p* = 0.628). Regarding metabolic outcomes, the LDL-c levels were positively correlated with the TSH levels (r = 0.472, *p* < 0.001) and negatively correlated with the FT4 levels (r = −0.244, *p* = 0.050). Additionally, the TG levels were positively correlated with the TSH levels (r = 0.232, *p* = 0.050), but not with the FT4 levels (r = −0.145, *p* = 0.239). All other correlations between body composition parameters and metabolic outcomes were not statistically significant. Correlation analyses between thyroid function tests (TSH and FT4) and body composition parameters (BMI z-scores, ASMM z-scores, and MFR z-scores), as well as metabolic outcomes in the study cohort, are presented in Table 3.

Forty-eight of the patients had at least one metabolic syndrome component, excluding obesity: seven had elevated diastolic BP, twenty-five had elevated systolic BP, nine had hypertriglyceridemia, twenty-three had low HDL, and nineteen had elevated TG/HDL ratios. The MFR z-scores were negatively correlated with the TG levels (r = −0.454, *p* < 0.001) and the TG/HDL-c ratio (r = −0.459, *p* < 0.001). The ASMM z-scores were positively correlated with the TG levels (r = 0.274, *p* = 0.019) and the TG/HDL-c ratio (r = 0.307, *p* = 0.011). A negative correlation was found between the HDL-c levels and ASMM z-scores (r = −0.311, *p* = 0.010), while the HDL-c levels were positively correlated with the MFR z-scores (r = 0.395, *p* < 0.001). All other correlations were not significant. Correlation analyses between the body composition parameters and metabolic outcomes are presented in Table 4.

## 4. Discussion

In this observational study, we explored the interactions between weight status, body composition, and thyroid function in children and adolescents with thyroid disorders. While the mean anthropometric measurements indicated satisfactory linear growth across the study cohort, nearly one-third of our patients were classified as overweight or obese. Body composition assessments revealed a profile of reduced appendicular skeletal muscle mass and muscle-to-fat ratio z-scores. Our findings indicated significant differences in clinical and metabolic characteristics across thyroid disorder groups. Notably, the TSH and FT4 levels varied markedly, with patients diagnosed with congenital hypothyroidism exhibiting the highest TSH levels, while those with subclinical hypothyroidism demonstrated the lowest FT4 levels. Associations between body composition parameters and metabolic outcomes suggested that alterations in thyroid function may influence both adiposity and metabolic health in this population.

Evidence of normal linear growth in our pediatric patients with thyroid disorders is a sensitive indicator that optimal thyroid hormone balance is being maintained through effective medical management. Thyroid hormones, particularly thyroxine (T4) and triiodothyronine (T3), play a crucial role in growth during childhood and adolescence [25]. T3, through its receptors, TRα1 and TRβ1, exerts pleotropic effects on the skeleton, stimulating the proliferation of chondrocytes and osteoblasts, which promote elongation and maturation of the bones [26]. Regular monitoring of the child’s growth rate, taking into account sex, age, and pubertal stage, is essential for assessing the appropriateness of the prescribed treatment for thyroid disorders and ensuring adherence to the treatment regimen. Achieving a height z-score near the genetic potential is another indicator that the medical treatment is effectively maintaining a euthyroid condition. Fortunately, the patients in our study had a mean delta height of zero, indicating that their current height aligned with the expected range based upon parental height.

Weight status, measured by the BMI z-score, is another important indicator of thyroid hormone balance, since untreated hypothyroidism may lead to overweight while hyperthyroidism may contribute to weight loss [27,28]. Proper long-term treatment and maintenance of the hormonal balance may help achieve a normalized weight status and prevent excessive adiposity [29]. Furthermore, alterations in thyroid function may be associated with obesity, including increased TSH and active triiodothyronine (T3) levels, without significant changes in T4. These findings suggest a disrupted energy balance and hormone resistance in obesity [30]. Notably, nearly one-third of our cohort was classified as overweight or obese, despite receiving treatment for their thyroid disorders. The positive interaction we observed between the patients’ BMI z-scores and their TSH levels points towards a link between TSH levels and increased levels of adiposity [31]. This finding is consistent with previous reports that highlighted the supraphysiologic increase in TSH in cases of overweight [2,28,32]. Excessive weight gain can disrupt the thyroid hormone balance, complicating the process of adjusting treatment for the underlying medical condition.

Furthermore, the mean muscle-to-fat ratio z-score was below zero, indicating an overall compromised body composition. A negative correlation was found between our patients’ MFR z-scores and their TSH levels, suggesting an interaction between weight status, MFR, and thyroid hormone homeostasis. Notably, the muscle component, as measured by ASMM z-scores, was associated with the FT4 levels but not with TSH. This relationship between body composition components and thyroid function has been described by others, particularly in cases of new-onset hypothyroidism [5]. Since the patients in our cross-sectional study have been regularly followed and treated in our endocrine clinic and their median thyroid function tests were within the normal range, the relatively high BMI z-scores and low MFR z-scores may stem from multiple factors, including lifestyle habits.

It is noteworthy that forty-eight of our study patients (32.7%) showed evidence of metabolic syndrome components. The TSH levels were associated with the lipoprotein levels, indicating that altered thyroid function can influence the low-density lipoprotein cholesterol (LDL-c) and triglyceride levels, thereby increasing the risk for cardiovascular and cerebrovascular diseases [6]. The interaction between the TSH and LDL-c levels observed in this study corroborates with the existing evidence that hypothyroidism is linked to elevated LDL-c levels [7]. This association underscores the role of thyroid hormones in lipid metabolism, where decreased thyroid function leads to a reduced clearance of LDL-c from the circulation, thereby increasing the cardiovascular risk [8]. The correlation between FT4 and LDL-c was not significant, indicating a complex interaction between thyroid hormones and lipoprotein levels in which T3 may have a key role. The interaction between the TSH and TG levels reinforces the metabolic impact of hypothyroidism, where underdiagnosed and untreated hypothyroidism can lead to hypertriglyceridemia. Our patients’ MFR z-scores were linked to their TG and TG/HDL-c ratios, indicating that thyroid dysfunction may affect lipoprotein levels through multiple mechanisms, including those employed in the determination of body composition.

Our findings highlight the importance of closely monitoring and managing a thyroid hormone balance and lipid profile to mitigate long-term metabolic risks in these young patients. Interestingly, the blood pressure percentiles were not linked to the thyroid function tests. We had previously reported that body composition parameters, specifically, the MFR z-scores, can predict the risk for early-onset metabolic syndrome components [33], and similar associations were demonstrated in the current cohort of children and adolescents with thyroid disorders.

This study has several limitations. The main limitation is its retrospective design, which limits our ability to establish causality between exposures and outcomes. The absence of a longitudinal design prevented the determination of whether a longer thyroid disorder duration is related to changes in body composition. Additionally, we did not include questionnaires to assess nutritional, behavioral, or psychological factors that could affect weight status and body composition. Another limitation of the study is the lack of comparison between the cohort’s data and that of sex- and age-matched controls without thyroid disorders. Missing data may have also introduced selection bias. As a tertiary referral center, our findings may not be fully representative of the broader pediatric population with thyroid disorders in the country or internationally. One of the major strengths of our study is the consistency in medical evaluation and care provided by the same multi-disciplinary team. This included body composition assessments, anthropometric measurements, blood pressure monitoring, and pubertal evaluations, all performed by trained medical staff. Another strength is the use of sex- and age-adjusted z-scores and percentiles, which enhanced the comparability and interpretation of the data.

## 5. Conclusions

To the best of our knowledge, this is the first study to investigate muscle-to-fat ratio z-scores in pediatric patients with thyroid disorders. While the cohort exhibited generally favorable anthropometric outcomes, we identified a concerning rate of obesity prevalence and an unfavorable body composition profile, along with an association between body composition parameters and metabolic syndrome components. Given the interaction between body composition parameters and thyroid function tests, efforts should be made to implement strategies that optimize lifestyle habits to improve body composition in young patients with thyroid disorders.

## Figures and Tables

**Figure 1 jcm-14-01255-f001:**
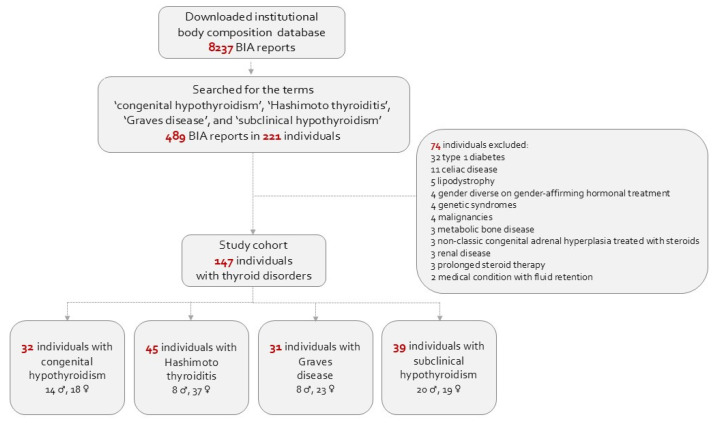
Flowchart of the study cohort.

**Table 1 jcm-14-01255-t001:** Sociodemographic and perinatal characteristics of the study cohort.

	Congenital Hypothyroidism, n = 32	Hashimoto Thyroiditis, n = 45	Graves’ Disease, n = 31	Subclinical Hypothyroidism, n = 39	*p* Value
Sex, n (%)					
Males (%)	14/32 (43.8) ^ab^	8/45 (17.8) ^c^	8/31 (25.8) ^bc^	20/39 (51.3) ^a^	**0.005**
Females (%)	18/32 (56.2)	37/45 (82.2)	23/31 (74.2)	19/39 (48.7)	
Socioeconomic position					
Cluster, median [IQR]	8 [6, 9]	8 [7, 9]	8 [6, 9]	8 [7, 9]	0.445
Index, median [IQR]	1.17 [0.23, 173]	1.469 [0.571, 1.805]	1.143 [0.559, 1.776]	1.259 [0.786, 1.769]	0.562
Low (1–4)	6/32 (18.8)	2/45 (4.4)	3/31 (9.7)	2/39 (5.1)	0.431
Medium (5–7)	8/32 (25.0)	13/45 (28.9)	10/31 (23.2)	12/39 (30.8)	
High (8–10)	18/32 (56.2)	30/45 (66.6)	18/31 (58.1)	25/39 (64.1)	
Children in the family					
Number	2 [2, 3]	3 [2, 3]	2 [2, 3]	3 [2, 3]	0.783
Birth order	2 [1, 3]	2 [1, 2]	2 [1, 2]	2 [1, 2]	0.677
Conception, n (%)					
Spontaneous	29/31 (93.5)	38/42 (90.5)	28/28 (100)	31/36 (86.1)	0.330
IVF	2/31 (6.5)	4/42 (9.5)	0 (0)	4/36 (11.1)	
Adopted	0 (0)	0 (0)	0 (0)	1/36 (2.8)	
Maternal conditions, n (%)					
GDM	1/31 (3.2)	5/42 (11.9)	2/28 (7.1)	0/36 (0)	0.307
Mode of delivery, n (%)					
Spontaneous vaginal	25/30 (83.3)	36/42 (85.7)	23/27 (85.2)	23/34 (67.6)	0.296
Elective C-section	3/30 (10.0)	5/42 (11.9)	3/27 (11.1)	10/34 (29.4)	
Urgent C-section	2/30 (6.7)	1/42 (2.4)	1/27 (3.7)	1/34 (2.9)	
Gestational age classifications, n (%)					
GA, weeks	40 [38, 40]	40 [38, 40]	40 [38, 40]	38.5 [37.3, 40]	0.439
Preterm, <37 wks	16/32 (50.0)	16/45 (35.5)	24/31 (77.4)	15/39 (38.5)	0.132
Term, 38–42 wks	3/32 (9.4)	3/45 (6.7)	1/31 (3.2)	2/39 (5.1)	
Postterm, ≥42 wks	2/32 (6.2)	3/45 (6.7)	1/31 (3.2)	7/39 (17.9)	
Birth parameters and categories					
Birth weight, grams	2930 [2732, 3215]	3282 [3006, 3550]	3100 [3000, 3300]	3160 [2800, 3390]	0.072
Birth weight, z-score	−0.49 ± 0.88	0.07 ± 0.90	−0.09 ± 0.78	−0.23 ± 0.90	0.061
SGA, n (%)	3/32 (9.4)	2/45 (4.4)	1/31 (3.2)	3/39 (7.7)	0.911
AGA, n (%)	27/32 (84.4)	40/45 (88.9)	29/31 (93.6)	33/39 (84.6)	
LGA, n (%)	2/32 (6.2)	3/45 (6.7)	1/31 (3.2)	3/39 (7.7)	

For quantitative data with normal distribution, expressed as mean ± standard deviation, the one-way ANOVA test was performed with the post hoc Mann–Whitney Tukey’s test to compare between groups. For quantitative data with skewed distribution, expressed as the median [interquartile range], an independent sample Kruskal–Wallis test was performed to compare between groups, with adjustment by Bonferroni correction for multiple post hoc tests. Categorical data were expressed as number and (percent), and the Chi-square test or Fischer exact test was performed for comparison. The values with different superscript letters (a, b, and c) in a column are significantly different from each other in pairwise comparisons (*p* ≤ 0.05). Mode of conception was reported for 93% of the cohort, while mode of delivery was documented for 92%. Abbreviations: IQR, interquartile range; IVF, in vitro fertilization; GDM, gestational diabetes mellitus; GA, gestational age; SGA, small for gestational age; AGA, appropriate for gestational age; LGA, large for gestational age. A *p*-value of ≤0.05 was considered significant. Bold indicates significant.

**Table 2 jcm-14-01255-t002:** Clinical and laboratory characteristics of the study cohort.

	Congenital Hypothyroidism n = 32	Hashimoto Thyroiditis n = 45	Grave’s Disease n = 31	Subclinical Hypothyroidism n = 39	*p* Value
Current age, years	9.4 [7.2, 11.7] ^a^	15.7 [11.4, 17.2] ^bc^	16.5 [14.1, 18.3] ^c^	13.5 [11.1, 16.4] ^b^	**<0.001**
Pubertal status, n (%)					
Prepubertal (Tanner 1)	21 (66) ^b^	9 (20) ^a^	2 (6.5) ^a^	8 (20.5) ^a^	**<0.001**
In puberty (Tanner 2–4)	8 (25)	9 (20)	4 (13)	15 (38.5)	
Fully pubertal (Tanner 5)	3 (9)	27 (30)	25 (80.5)	16 (41)	
Anthropometric measurements					
Height, z-score	−0.30 ± 1.21	−0.90 ± 0.86	0.22 ± 1.00	0.16 ± 0.98	0.076
MPHt, z-score	−0.24 ± 0.80	−0.02 ± 0.68	0.05 ± 0.89	−0.20 ± 0.96	0.551
Delta height, z-score	−0.13 [−1.00, 0.67]	0.07 [−0.72, 0.49]	0.23 [−0.31, 0.55]	0.37 [−0.35, 0.98]	0.064
Weight, z-score	−0.03 ± 1.50	0.52 ± 1.23	0.12 ± 0.89	0.43 ± 1.83	0.112
BMI, z-score	0.26 [−0.24, 1.20] ^ab^	1.07 [−0.05, 1.39] ^a^	−0.09 [−0.58, 0.49] ^b^	0.61 [−0.25, 1.36] ^ab^	**0.010**
Body composition components					
Fat percentage	24.3 ± 6.2 ^a^	29.5 ± 7.7 ^b^	24.5 ± 7.7 ^a^	25.0 ± 6.5 ^ab^	**0.002**
Truncal fat percentage	19.0 ± 6.0 ^a^	24.1 ± 7.4 ^b^	19.3 ± 7.6 ^a^	19.7 ± 6.0 ^a^	**0.001**
ASMM, z-score	−0.65 ± 1.00	−0.03 ± 1.09	−0.18 ± 0.68	−0.24 ± 1.03	0.058
MFR, z-score	−0.66 ± 0.73 ^a^	−0.77 ± 0.76 ^a^	−0.12 ± 1.12 ^b^	−0.72 ± 0.75 ^a^	**0.003**
Thyroid function tests at body composition assessment					
TSH, mIU/L	6.69 [3.54, 11.38] ^a^	5.97 [3.08, 9.80] ^a^	1.50 [0.89, 3.41] ^b^	4.57 [2.44, 7.29] ^a^	**<0.001**
Free T4, ng/dL	1.28 [1.17, 1.35] ^b^	1.18 [0.98, 1.24] ^a^	1.13 [1.02, 1.30] ^ab^	1.10 [1.02, 1.22] ^a^	**<0.001**
Lipid profile at body composition assessment					
Cholesterol, mg/dL	172.3 ± 28.7 ^a^	176.9 ± 31.5 ^a^	142.3 ± 27.5 ^b^	167.2 ± 24.0 ^a^	**<0.001**
Triglycerides, mg/dL	61.8 ± 18.5 ^a^	101.0 ± 29.5 ^b^	71.9 ± 22.3 ^ab^	70.8 ± 20.9 ^ab^	**<0.001**
HDL-c, mg/dL	61.2 ± 12.5 ^a^	53.9 ± 11.6 ^ab^	49.6 ± 12.9 ^b^	59.6 ± 15.2 ^ab^	**0.044**
LDL-c, mg/dL	99.9 ± 22.1 ^a^	102.9 ± 24.0 ^a^	79.7 ± 25.2 ^b^	92.9 ± 20.2 ^ab^	**<0.001**
Non-HDL-c, mg/dL	113.9 ± 21.6 ^a^	122.7 ± 23.3 ^a^	95.1 ± 27.4 ^b^	107.7 ± 20.2 ^ab^	**<0.001**
TG:HDL	1.11 ± 0.45 ^a^	1.90 ± 0.91 ^b^	1.66 ± 0.68 ^ab^	1.39 ± 0.89 ^ab^	**<0.001**

For quantitative data with normal distribution, expressed as mean ± standard deviation, the one-way ANOVA test was performed with the post hoc Mann–Whitney two-key test to compare between groups. For quantitative data with skewed distribution, expressed as the median [interquartile range], an independent sample Kruskal–Wallis test was performed to compare between groups, with adjustment by Bonferroni correction for multiple post hoc tests. Categorical data were expressed as the number and (percent), and the chi-square test or Fischer exact test was performed for comparison. The values with different superscript letters (a, b, and c) in a column are significantly different from each other in pairwise comparisons (*p* ≤ 0.05). Abbreviations: MPHt, mid-parental height; BMI, body mass index; ASMM, appendicular skeletal muscle mass; MFR, muscle-to-fat ratio. HDL-c, high-density lipoprotein cholesterol; LDL-c, low-density lipoprotein cholesterol; Non-HDL-c, non-high-density lipoprotein cholesterol, TG, triglycerides. A *p*-value of ≤0.05 was considered significant. Bold indicates significant.

**Table 3 jcm-14-01255-t003:** Correlations between the thyroid function tests, body composition parameters, and metabolic outcomes.

	TSH		FT4	
	r	*p*	r	*p*
BMI, z-score	0.238	**0.005**	−0.165	0.057
ASMM, z-score	0.091	0.291	−0.255	**0.003**
MFR, z-score	−0.215	**0.012**	0.042	0.628
Metabolic outcomes				
Systolic blood pressure, %	0.032	0.708	0.122	0.161
Diastolic blood pressure, %	−0.046	0.592	0.113	0.194
Low-density lipoprotein cholesterol	0.472	**<0.001**	−0.244	0.052
High-density lipoprotein cholesterol	0.084	0.500	0.036	0.779
Triglycerides	0.232	**0.050**	−0.145	0.239
TG/HDL-c	0.077	0.536	−0.103	0.414

Spearman’s correlation coefficient statistical tests were applied. Abbreviations: BMI, body mass index; ASMM, appendicular skeletal muscle mass; MFR, muscle-to-fat ratio; TG/HDL-c, triglyceride to high-density lipoprotein cholesterol ratio. Bold indicates significant.

**Table 4 jcm-14-01255-t004:** Correlations between the body composition parameters and metabolic outcomes.

	ASMM, z-Score		MFR, z-Score	
	r	*p*	r	*p*
Systolic blood pressure, **%**	0.104	0.211	−0.161	0.052
Diastolic blood pressure, **%**	−0.029	0.724	−0.179	**0.030**
Low-density lipoprotein cholesterol	0.066	0.595	−0.217	0.078
High-density lipoprotein cholesterol	−0.311	**0.010**	0.395	**<0.001**
Triglycerides	0.274	**0.019**	−0.454	**<0.001**
TG/HDL-c	0.307	**0.011**	−0.459	**<0.001**

Spearman’s correlation coefficient statistical tests were applied. Abbreviations: ASMM, appendicular skeletal muscle mass; MFR, muscle-to-fat ratio; TG/HDL-c, triglyceride to high-density lipoprotein cholesterol ratio. Bold indicates significant.

## Data Availability

The data used to support the findings of this study are available from the corresponding author upon reasonable request.

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
