# Peer review of "Exploring the Muscle-to-Fat Ratio of Pediatric Patients with Thyroid Disorders and Its Interaction with Thyroid Function and Metabolic Syndrome Components"

_jcm, 2025, doi:10.3390/jcm14041255_

Round 1

Reviewer 1 Report

Comments and Suggestions for Authors

After reviewing the manuscript, the following observations and suggestions were made. 

In the title 

It is suggested to delete, change or modify the word association. The present study shows data analysis and results in its correlation form, the statistical correlation is not an “association” of the variables. It does not imply causality. 

In the abstract 

Avoid the use of abbreviations, it is suggested to place the full name of the abbreviations and describe them better in the materials and methods section, this will save space and will also avoid distraction from the reading of this section. 

In Keywords

Suggested keywords: correlation; Metabolic syndrome; Thyroid function. 

Introduction 

In this section, the authors begin by mentioning the importance of the normal function of the thyroid gland, however, in no section of the document are the average values or acceptable values of normality in the pediatric group, or in the different age groups that comprise this pediatric analysis group, it is suggested to give a description in relation to these values of normality in order to understand their importance and to understand the changes that can occur after the disruption of the thyroid function. 

There is no general or simple description of what happens with the levels of thyroid hormones in the presence of disruptions in the thyroid gland. Are they produced more or less or are they nonexistent, given the variety of disruptions that can occur.

It is suggested to review this section and to complement with updated information the missing part for a better understanding of the objective of the study. 

In materials and methods 

In the “study design” section, it is mentioned that children and adolescents were included, but the title of the document indicates “pediatric patients”; here is where it becomes relevant to designate the definition of pediatric age. 

It is suggested to indicate in more detail the part of the sample selection criteria, with the inclusion, exclusion and elimination criteria. It is mentioned that patients with thyroid disorders were included, what about those patients with congenital hypothyroidism or Hashimoto's disease who are still taking levothyroxine in their treatment? Were they included? Were they excluded? Were they eliminated? The same with the other patients who included in their treatment the intake of thyroid hormones or some pharmacological supplement to compensate the over or under production of thyroid hormones, it is suggested to detail that part in this section. 

Why were patients included from the age of 5 years and not before?  Because patients were included up to 19 years of age and not up to 21 years of age, as established by the AAP (American Academy of Pediatrics) late pediatric age classification of 21 years. 

In the last line of the “study design” section it is mentioned: The study population flowchart is presented in Figure 1; however, the title of the figure is Flowchart of the study cohort. The type of study that was carried out should be specified; it is not the same as an observational, cross-sectional, case-control, cohort, or clinical trial study. It cannot be a cohort study because no interventions with their designated follow-up over time were performed. Only one measurement was made at a given time, this being a cross-sectional study. It cannot be considered a cohort study when the variables analyzed were not measured in their entirety, for example, the data collected on perinatal characteristics are not the same as the variables used for the other analyses (Table 2 and 3). It can be observed that there is a lack of understanding of these definitions and applications, and it is strongly suggested that this section be reviewed.

In the section on the definition of the study variables there is no definition for the variables of 'congenital hypothyroidism', 'Hashimoto thyroiditis', 'Graves' disease', and 'subclinical hypothyroidism', as it is known that this diagnosis is still correct or as it is certain that it has not changed, this part should be mentioned as it is crucial as a basis for the study, especially in this type of thyroid disorders that need different and continuous treatments. 

In the statistical analysis section, certain types of variables are specified but others (categorical) are not specified, it is suggested to pay attention to this part. 

In the results, the whole thyroid profile of the participants is not appreciated, this is crucial since the manuscript bases all its interpretation on thyroid disorders, only THS and T4 are appreciated, how is this adequate for the type of inference they are trying to address?

The authors indicate limitations to their study, this is something worth mentioning and is of utmost importance, however, it is also an unfavorable point in the sense that they know the weaknesses of their study but no approaches are observed to minimize their effects on the analysis performed, attention should be paid to this part in order to highlight in a better way all the work done. 

In the conclusions 

The authors emphasize “this is the first study to investigate muscle-to-fat ratio z-scores in pediatric patients with thyroid disorders”, this section differs from what is shown in the final section of the introduction, it is suggested that the authors identify their general and specific objectives in order to be coherent in the wording, and in the scope that their work may have.   The discussion and conclusion section will undoubtedly change once the points of this round of review have been reviewed. 

In the tables, attention should be paid to the titles, table captions and images. The type of statistical analysis carried out should be included and the p-value should be standardized in all tables, not only the P. 

Table 1 shows p-values according to the variables used, but in the section of the manuscript under discussion it is not given an adequate description, how important is it?

The same in table 2, the type of statistical analysis implemented in that table should be placed. 

The same for table 3, in addition it should be specified what is meant by TSH and T4, that correlation is given when it is more than or less than, both TSH and T4, in this type of correlations is more understandable a graph, or in any case if you prefer to use a table should be placed the values to which the correlation alludes, not only the name of the variable but its continuous data. 

There are other points that can be addressed but we really need to pay attention to the basics in order to understand the direction of the manuscript. 

Author Response

Response: We thank the reviewer for the time and effort expended in providing valuable feedback to enhance the quality of our manuscript. We have carefully revised the manuscript as advised, and our detailed, point-by-point responses are listed below.

In the title:

It is suggested to delete, change or modify the word association. The present study shows data analysis and results in its correlation form, the statistical correlation is not an “association” of the variables. It does not imply causality.

Response: The title was revised and now reads “Exploring the muscle-to-fat ratio of pediatric patients with thyroid disorders and its interactions with thyroid function and metabolic syndrome components”.

In the abstract:

Avoid the use of abbreviations, it is suggested to place the full name of the abbreviations and describe them better in the materials and methods section, this will save space and will also avoid distraction from the reading of this section.

Response: As suggested, all of the abbreviations have now been written out in full.

In Keywords:

Suggested keywords: correlation; Metabolic syndrome; Thyroid function.

Response: the keywords ‘thyroid function’ and ‘lipid profile’ were added, with thanks. 

Introduction:

In this section, the authors begin by mentioning the importance of the normal function of the thyroid gland, however, in no section of the document are the average values or acceptable values of normality in the pediatric group, or in the different age groups that comprise this pediatric analysis group, it is suggested to give a description in relation to these values of normality in order to understand their importance and to understand the changes that can occur after the disruption of the thyroid function.

Response: As advised, we have added the following to the revised methods section: "All patients with hypo- or hyperthyroidism were treated to achieve clinical and laboratory euthyroid condition. Laboratory results were interpreted according to age-appropriate reference ranges (provided by the manufacturer) as follows. TSH (mIU/mL): from birth to 1 month = 0.64-10; from 1 to 6 months = 0.64-7; from 6 months to 1 year = 0.64-6.27; from 1 to 18 years = 0.51-4.94; from 18 years onwards = 0.40-4.70. FT4 (ng/dL): from birth to 1 month = 0.66-2.37; 1 month to 1 year = 0.71-1.97; from 1 year onwards: 0.70-1.80."

There is no general or simple description of what happens with the levels of thyroid hormones in the presence of disruptions in the thyroid gland. Are they produced more or less or are they nonexistent, given the variety of disruptions that can occur. It is suggested to review this section and to complement with updated information the missing part for a better understanding of the objective of the study.

Response: As suggested, the revised methods section clarifies that the study patients are already treated based upon their diagnosis and with the goal of achieving and maintaining clinical and laboratory euthyroidism.

In materials and methods

In the “study design” section, it is mentioned that children and adolescents were included, but the title of the document indicates “pediatric patients”; here is where it becomes relevant to designate the definition of pediatric age.

Why were patients included from the age of 5 years and not before?  Because patients were included up to 19 years of age and not up to 21 years of age, as established by the AAP (American Academy of Pediatrics) late pediatric age classification of 21 years.

Response: The reviewer’s comments are well taken. All of the patients diagnosed with thyroid disorders who were treated in our clinic underwent body composition assessment during their visits, but bioimpedance analysis is available only for patients aged 5 years and older, while those aged 20 years and above are transferred to the adult endocrinology clinic.

It is suggested to indicate in more detail the part of the sample selection criteria, with the inclusion, exclusion and elimination criteria. It is mentioned that patients with thyroid disorders were included, what about those patients with congenital hypothyroidism or Hashimoto's disease who are still taking levothyroxine in their treatment? Were they included? Were they excluded? Were they eliminated? The same with the other patients who included in their treatment the intake of thyroid hormones or some pharmacological supplement to compensate the over or under production of thyroid hormones, it is suggested to detail that part in this section.

Response: Our thanks for bringing our attention to this lack of clarification. We now state that the patients included in this cross-sectional study are already receiving treatment tailored to their medical diagnosis, with adjustments made based upon their clinical and laboratory findings. The additional text reads: “All patients with hypo- or hyperthyroidism weare treated to achieve clinical and laboratory euthyroid condition."

In the last line of the “study design” section it is mentioned: The study population flowchart is presented in Figure 1; however, the title of the figure is Flowchart of the study cohort. The type of study that was carried out should be specified; it is not the same as an observational, cross-sectional, case-control, cohort, or clinical trial study. It cannot be a cohort study because no interventions with their designated follow-up over time were performed. Only one measurement was made at a given time, this being a cross-sectional study. It cannot be considered a cohort study when the variables analyzed were not measured in their entirety, for example, the data collected on perinatal characteristics are not the same as the variables used for the other analyses (Table 2 and 3). It can be observed that there is a lack of understanding of these definitions and applications, and it is strongly suggested that this section be reviewed.

Response: We have now clarified that this was a “retrospective, observational cross-sectional study” wherever relevant throughout the revised text. The perinatal information was recalled during the anamnesis and recorded in the patients' medical files.

In the section on the definition of the study variables there is no definition for the variables of 'congenital hypothyroidism', 'Hashimoto thyroiditis', 'Graves' disease', and 'subclinical hypothyroidism', as it is known that this diagnosis is still correct or as it is certain that it has not changed, this part should be mentioned as it is crucial as a basis for the study, especially in this type of thyroid disorders that need different and continuous treatments.

Response: As advised, the revised Methods section includes (and references) the widely accepted criteria for diagnosing the various thyroid disorders discussed in this study. Specifically, references 10-14.

In the statistical analysis section, certain types of variables are specified but others (categorical) are not specified, it is suggested to pay attention to this part.

Response: All of the variables analyzed as “categorical” have been listed in the revised statistical analysis section, with thanks.

In the results, the whole thyroid profile of the participants is not appreciated, this is crucial since the manuscript bases all its interpretation on thyroid disorders, only THS and T4 are appreciated, how is this adequate for the type of inference they are trying to address?

Response: The routine laboratory follow-up for all patients with thyroid disorders included thyroid function tests (TSH and FT4), which are used to adjust medication doses for those requiring pharmacotherapy. Autoantibodies are utilized for the diagnosis of autoimmune thyroid disorders but are not regularly obtained as part of standard care.

The authors indicate limitations to their study, this is something worth mentioning and is of utmost importance, however, it is also an unfavorable point in the sense that they know the weaknesses of their study but no approaches are observed to minimize their effects on the analysis performed, attention should be paid to this part in order to highlight in a better way all the work done.

Response: This is a retrospective cross-sectional study aimed at exploring the body composition components of treated patients with various thyroid disorders. We recognize its descriptive nature, but the information presented in this manuscript has not been published previously and we believe it will be of interest to pediatric and pediatric endocrinology readers.

In the conclusions

The authors emphasize “this is the first study to investigate muscle-to-fat ratio z-scores in pediatric patients with thyroid disorders”, this section differs from what is shown in the final section of the introduction, it is suggested that the authors identify their general and specific objectives in order to be coherent in the wording, and in the scope that their work may have.   The discussion and conclusion section will undoubtedly change once the points of this round of review have been reviewed.

Response: As suggested, the objective outlined in the Introduction has been revised for consistency with the wording in the Conclusions: “In this study, we explored the body composition components and examined their interaction with thyroid function and metabolic outcomes in children and adolescents treated and followed for various thyroid disorders".

In the tables, attention should be paid to the titles, table captions and images. The type of statistical analysis carried out should be included and the p-value should be standardized in all tables, not only the P.

Response: Done, with thanks.

Table 1 shows p-values according to the variables used, but in the section of the manuscript under discussion it is not given an adequate description, how important is it?

Response: The data presented in the table on the sociodemographic and perinatal characteristics of the study groups aim to provide readers with an overview of the study cohort.

The same in table 2, the type of statistical analysis implemented in that table should be placed.

Response: All of the statistical analyses applied for comparison were added to the revised tables.

The same for table 3, in addition it should be specified what is meant by TSH and T4, that correlation is given when it is more than or less than, both TSH and T4, in this type of correlations is more understandable a graph, or in any case if you prefer to use a table should be placed the values to which the correlation alludes, not only the name of the variable but its continuous data.

Response: Please note that the values of thyroid function tests (TSH, FT4) are presented in Table 2. Also, Table 3 presents the association between the thyroid function tests and body composition parameters, blood pressure percentiles and levels of lipoproteins.

There are other points that can be addressed but we really need to pay attention to the basics in order to understand the direction of the manuscript.

Reviewer 2 Report

Comments and Suggestions for Authors

Dear Authors,

Below I include some minor and major comments and suggestions that should be taken into consideration before the paper can be processed further:

-       I am not sure whether the abstract of your paper meets the standards of the MDPI and JCM specifically – in this form it seems to be too long in words. Please check the guidelines for the authors and check the maximum number of words the abstract can have and correct accordingly

-       Also, the citation style does not meet the standards of JCM. Please write the guidelines for the authors first and prepare the manuscript according to them before you submit a paper anywhere

-       Figure 1 is of very low quality. I am not even sure what is included in this picture. Please provide it in a better quality

-       Table 1 should be a little bit wider. Also, I highly recommend including it in the text, preferably in the results section to make this presentation clearer for the readers. Further, all the explanations below the table does not necessarily need to be included IN the table – you can add it just below the table

-       Please check the manuscript once again in terms of English since there are some grammatical errors that should be corrected before the paper can be processed further

Author Response

Response: Our thanks to the reviewer for the time and effort expended on our behalf to enhance the presentation of our investigation. We have carefully considered the reviewer’s suggestions and have made the suggested revisions as listed below.

-       I am not sure whether the abstract of your paper meets the standards of the MDPI and JCM specifically – in this form it seems to be too long in words. Please check the guidelines for the authors and check the maximum number of words the abstract can have and correct accordingly

-       Also, the citation style does not meet the standards of JCM. Please write the guidelines for the authors first and prepare the manuscript according to them before you submit a paper anywhere

Response: This has been corrected.

 Figure 1 is of very low quality. I am not even sure what is included in this picture. Please provide it in a better quality

Response: The quality of Figure 1 had now been enhanced.

-       Table 1 should be a little bit wider. Also, I highly recommend including it in the text, preferably in the results section to make this presentation clearer for the readers. Further, all the explanations below the table does not necessarily need to be included IN the table – you can add it just below the table

Table 1 will appear in print after first mention according to the JCM format.

Response: The explanations have been moved to appear as footnotes, as advised.

-       Please check the manuscript once again in terms of English since there are some grammatical errors that should be corrected before the paper can be processed further

Response: We carefully checked for errors as advised.

Reviewer 3 Report

Comments and Suggestions for Authors

This retrospective study by Brener et al examined body composition and serum lipids in 147 children and adolescents with various thyroid disorders who underwent bioimpedance analysis.  The paper is primarily descriptive. Overweight was common and skeletal muscle mass was slightly reduced especially among those with congenital hypothyroidism. TSH was statistically, although weakly, positively correlated with BMI. The objectives of the study are clearly presented, and the paper is well written.  Much of the findings are confirmatory.

Comments

1.        Line 74. When were Graves’ disease patients studied in relation to hyperthyroidism or post-hyperthyroidism treatment?

2.        Line 74. Was there overlap between “Hashimoto thyroiditis” and “subclinical hypothyroidism”?

3.        Lines 167 and 195. The lack of an ethnically matched control group is a substantial limitation in understanding the prevalence findings. This is a limitation of the research plan.

4.        Line 220 and 249 imply that subjects were receiving thyroid hormone treatments at the time of testing and BIA analysis. Were subjects receiving LT4? T3?  Details are needed (line 98). The duration of treatment may influence the results and interpretation of all between group differences (line 212); furhter discussion is needed. 

5.        Table 1. It is not clear which variables are compared statistically. The groups with p-value analysis should be specified with superscripts or in the Legend. 

6.        Table 3. Given the positive correlation between TSH and BMI, a mention of the relationship of TSH to fat mass would be of interest.

7.        Childhood BMI is often related to maternal or mid-parental BMI. This analysis would be of interest.

8.         Line 147. That the correlation was inverse and that the MFR was lowest in those with Graves’ disease should be noted.

9.        Line 251. Median TSH levels of 6.69 and 5.97 mIU/L (Table 2) exceed most reference ranges. The assays used and reference ranges were not described.

10.   The present findings confirm previously published results. Dahl et al (J Clin Res Ped Endocrinol 1:8,2017) and Jin (J Ped Child Health 54:975, 2018) reported that subclinical (mild) hypothyroidism was more common in adolescents with obesity while Lee et al (Endocrine 65:606,2019) reported that the risk for abdominal obesity was increased among adolescents with mild hypothyroidism. Jin also reported higher cholesterol and triglyceride levels in adolescents with mild hypothyroidism.  Livadas et al (Horm Metab Res 39:524,2007) found that the BMI of children with congenital hypothyroidism normalizes as children grow older.

These similar studies should be included and compared to the current results. In addition, these results favor the designation “mild” rather than “subclinical” hypothyroidism since obesity and dyslipidemia are not subclinical.

Minor comments.

11.   Line 15. “includes’” seems more accurate than “consists of”.

12.   Line 139. “Chi-squared test exact test” or Chi-square test”?

13.   Line 146. The number of females/males should be added.

14.   Line 148. SEP cluster and SEP index should be defined.

15.   Table 1. Column 4, Graves’ disease. There is also a typographical error in column 1 at the bottom of the table.

Author Response

This retrospective study by Brener et al examined body composition and serum lipids in 147 children and adolescents with various thyroid disorders who underwent bioimpedance analysis.  The paper is primarily descriptive. Overweight was common and skeletal muscle mass was slightly reduced especially among those with congenital hypothyroidism. TSH was statistically, although weakly, positively correlated with BMI. The objectives of the study are clearly presented, and the paper is well written.  Much of the findings are confirmatory.

Response: Many thanks for the kind remarks and for the time and effort in providing us with such constructive feedback to enhance the quality of our work.

Comments

  1. Line 74. When were Graves’ disease patients studied in relation to hyperthyroidism or post-hyperthyroidism treatment?

Response: The Graves’ disease patients had already been receiving treatment and were under ongoing surveillance and treatment adjustments in the endocrine clinic. This clarification has been added to the revised Methods section, with thanks. "All patients with hypo- or hyperthyroidism were treated to achieve clinical and laboratory euthyroid condition."   

  1. Line 74. Was there overlap between “Hashimoto thyroiditis” and “subclinical hypothyroidism”?

Response: In the revised Methods section, we have included a paragraph detailing the definitions of each thyroid disorder included in the study. Patients with Hashimoto thyroiditis are characterized by the presence of thyroid autoantibodies, whereas those with subclinical hypothyroidism exhibit only elevated TSH levels without additional clinical or laboratory abnormalities.

  1. Lines 167 and 195. The lack of an ethnically matched control group is a substantial limitation in understanding the prevalence findings. This is a limitation of the research plan.

Response: Our population is ethnically matched to the reference group presented the McCarthy article (McCarthy et al. Skeletal Muscle Mass Reference Curves for Children and Adolescents. Pediatr. Obes. 2014, 9, 249–259).

  1. Line 220 and 249 imply that subjects were receiving thyroid hormone treatments at the time of testing and BIA analysis. Were subjects receiving LT4? T3? Details are needed (line 98). The duration of treatment may influence the results and interpretation of all between group differences (line 212); furhter discussion is needed.

Response: Patients with hypothyroidism were treated with LT4. Since longitudinal data were not available, we could not report changes over time. The following was added to the revised limitations section: “The absence of longitudinal design prevented the determination of whether a longer thyroid disorder duration is related to changes in body composition”.  

  1. Table 1. It is not clear which variables are compared statistically. The groups with p-value analysis should be specified with superscripts or in the Legend.

Response: As advised, we have now added superscript letters (a, b) to birthweight values which were significantly different from each other in pairwise comparison (p ≤ 0.05).

  1. Table 3. Given the positive correlation between TSH and BMI, a mention of the relationship of TSH to fat mass would be of interest.

Response: The reviewer’s point is very well taken and we now added the following to the revised Discussion: “The positive interaction we observed between the patients' BMI z-scores and their TSH levels points towards a link between TSH levels and increased levels of adiposity."

7.Childhood BMI is often related to maternal or mid-parental BMI. This analysis would be of interest.

Response: Unfortunately, we were unable to analyze this relationship due to the lack of parental weight data in many medical records.

  1. 8. Line 147. That the correlation was inverse and that the MFR was lowest in those with Graves’ disease should be noted.

Response: The MFR z-score in patients with Graves’ disease was actually the highest.

  1. 9. Line 251. Median TSH levels of 6.69 and 5.97 mIU/L (Table 2) exceed most reference ranges. The assays used and reference ranges were not described.

Our thanks for pointing this out. The age-specific norms of TSH and FT4 levels have now been added to the revised Methods section: “Laboratory results were interpreted according to age-appropriate reference ranges (provided by the manufacturer) as follows. TSH (mIU/mL): from birth to 1 month = 0.64-10; from 1 to 6 months = 0.64-7; from 6 months to 1 year = 0.64-6.27; from 1 to 18 years = 0.51-4.94; from 18 years onwards = 0.40-4.70. FT4 (ng/dL): from birth to 1 month = 0.66-2.37; 1 month to 1 year = 0.71-1.97; from 1 year onwards: 0.70-1.80."

  1. 10. The present findings confirm previously published results. Dahl et al (J Clin Res Ped Endocrinol 1:8,2017) and Jin (J Ped Child Health 54:975, 2018) reported that subclinical (mild) hypothyroidism was more common in adolescents with obesity while Lee et al (Endocrine 65:606,2019) reported that the risk for abdominal obesity was increased among adolescents with mild hypothyroidism. Jin also reported higher cholesterol and triglyceride levels in adolescents with mild hypothyroidism. Livadas et al (Horm Metab Res 39:524,2007) found that the BMI of children with congenital hypothyroidism normalizes as children grow older.

Many thanks for providing us these highly relevant sources in the literature. We have incorporated some of them where appropriate in the revised Discussion section.

These similar studies should be included and compared to the current results. In addition, these results favor the designation “mild” rather than “subclinical” hypothyroidism since obesity and dyslipidemia are not subclinical.

Response: Subclinical hypothyroidism is a medical condition defined by isolated elevation of TSH levels and that it should not be mistaken for subclinical Hashimoto thyroiditis (which could be described as ‘mild’), which is distinguished by evidence of thyroid autoimmunity. For greater clarification, the well-accepted definition with the appropriate references have been added to the revised Methods section: “Subclinical hypothyroidism was defined as the sole elevation of TSH levels above the normal range for age [12], [13] in the presence of normal FT4 levels and absence of clinical manifestations."

Minor comments.

  1. 11. Line 15. “includes’” seems more accurate than “consists of”.

 Response: Changed.

  1. 12. Line 139. “Chi-squared test exact test” or Chi-square test”?

Response: Our thanks for spotting this error. The correct term "Chi-square test" has replaced all mention of Chi-squared test.

  1. 13. Line 146. The number of females/males should be added

Response: Added.

  1. 14. Line 148. SEP cluster and SEP index should be defined.

Response: Done.

  1. Table 1. Column 4, Graves’ disease. There is also a typographical error in column 1 at the bottom of the table.

Response: Corrected.

Round 2

Reviewer 2 Report

Comments and Suggestions for Authors

Dear Authors,

Thank you for correcting the manuscript according to my comments and suggestions.

I have no further comments regarding this paper

Regards

a Reviewer 

Author Response

We sincerely appreciate the time and effort invested by the reviewer in enhancing the presentation of our study. 

Reviewer 3 Report

Comments and Suggestions for Authors

The authors have explained most of my inquiries and revised the manuscript in accordance with my comments. However, a few issues remain.

3. The manuscript include the prevalence of metabolic syndrome and its components in the study population. A comparison with a nonthyroid disease control group would make these prevalence results more meaningful. The absence of these data is a limitation of the paper in its current form.

6. A positive correlation between TSH and BMI was found. Since BMI includes fat and muscle mass, and fat mass was measured by BIA, it is unclear why the authors have not analyzed the relationship between TSH and fat mass, as requested.

10. These published papers deal with the same subject matter as the present report. Their findings should be discussed. Not to discuss published findings is unprofessional.

Line 223 and Table 2 Legend. Anova should be ANOVA. Two key test is Tukey's test. 

Author Response

Reviewer 3:
Comments and Suggestions for Authors
The authors have explained most of my inquiries and revised the manuscript in accordance with my comments. However, a few issues remain.
Response: We thank the reviewer for the constructive feedback, which has greatly contributed to improving the quality of our manuscript.
3. The manuscript include the prevalence of metabolic syndrome and its components in the study population. A comparison with a nonthyroidal disease control group would make these prevalence results more meaningful. The absence of these data is a limitation of the paper in its current form.
Response: Thanks for your valuable feedback. Our results have been standardized, and z-scores were calculated based on accepted normative values. However, as data from sex- and age-matched controls without thyroid disorders was not used for comparison, we have now acknowledged this in the revised limitations section.
6. A positive correlation between TSH and BMI was found. Since BMI includes fat and muscle mass, and fat mass was measured by BIA, it is unclear why the authors have not analyzed the relationship between TSH and fat mass, as requested.
Response: We calculated correlations only for the anthropometric and body composition components for which z-scores were determined. Given that fat mass varies considerably by sex, age, and pubertal status, its correlation with TSH would be influenced by these factors.
10. These published papers deal with the same subject matter as the present report. Their findings should be discussed. Not to discuss published findings is unprofessional.
Response: This important comment is well taken. A summary of the main findings from the cited articles has now been added to the revised introduction, as follows: “Body weight and fat mass were associated with thyroid hormone concentrations in patients with both hypothyroidism and hyperthyroidism [4]. In men without thyroid disorders, thyroid hormone levels were associated with adiposity indices but not with TSH [5].”
Line 223 and Table 2 Legend. Anova should be ANOVA. Two key test is Tukey's test.
Response: fixed.